# *Scutellaria petiolata* Hemsl. ex Lace & Prain (Lamiaceae).: A New Insight in Biomedical Therapies

**DOI:** 10.3390/antiox11081446

**Published:** 2022-07-26

**Authors:** Sidra Mubin, Najeeb Ur Rehman, Waheed Murad, Muddaser Shah, Ahmed Al-Harrasi, Rabia Afza

**Affiliations:** 1Department of Botany, Hazara University Mansehra, Mansehra 21310, Pakistan; shahhu123@gmail.com; 2Natural and Medical Sciences Research Center, University of Nizwa, Birkat Al Mauz, Nizwa 616, Oman; najeeb@unizwa.edu.om; 3Department of Botany, Abdul Wali Khan University Mardan, Mardan 23200, Pakistan; waheedmurad@awkum.edu.pk

**Keywords:** *Scutellaria petiolata*, phytochemical, in vitro antibacterial, antifungal, antioxidant, in vivo anti-inflammatory, analgesic activities

## Abstract

The recent investigation was designed to explore *Scutellaria petiolata* Hemsl. ex Lace & Prain (Lamiaceae) whole plant in various extracts (methanol (SPM), dichloromethane (SPDCM), n-Hexane (SPNH), and aqueous (SPAQ) for a phytochemicals assessment, ESI-LC-MS chemical analysis, in vitro antimicrobials, and antioxidants and in vivo anti-inflammatory and analgesic potential. The qualitative detection shows that all the representative groups were present in the analyzed samples. The examined samples display the greatest amount of total flavonoid content (TFC, 78.2 ± 0.22 mg QE/mg) and total phenolic contents (TPC, 66.2 ± 0.33 mg GAE/g) in the SPM extract. The SPM extract proceeded to the ESI-LC-MS to identify the chemical constituents that presented nineteen bioactive ingredients, depicted for the first time from *S. petiolata* mainly contributed by flavonoids. The analyzed samples produced considerable capability to defy the microbes. The SPM extract was observed effective and offered an appreciable zone of inhibition (ZOI), 17.8 ± 0.04 mm against the bacterial strain *Salmonella**typhi* and 18.8 ± 0.04 mm against *Klebsiella pneumonia*. Moreover, the SPM extract also exhibited 19.4 ± 0.01 mm against the bacterial strains *Bacillus atrophaeus* and 18.8 ± 0.04 mm against *Bacillus subtilis* in comparison to the standard levofloxacin (Gram-negative) and erythromycin (Gram-positive) bacterial strains that displayed 23.6 ± 0.02 mm and 23.2 ± 0.05 mm ZOI, correspondingly. In addition to that, the SPD fraction was noticed efficiently against the fungal strains used with ZOI 19.07 ± 0.02 mm against *Aspergillus parasiticus* and 18.87 ± 0.04 mm against the *Aspergillus niger* as equated to the standard with 21.5 ± 0.02 mm ZOI. In the DPPH (2,2-diphenyl-1-picrylhydrazyl) analysis, the SPM extract had the maximum scavenging capacity with IC_50_ of 78.75 ± 0.19 µg/mL succeeded by the SPDCM fraction with an IC_50_ of 140.50 ± 0.20 µg/mL free radicals scavenging potential. Through the ABTS (2,2′-azino-bis (3-ethylbenzothiazoline-6-sulfonic acid) assay, the similar extract (SPM) presented an IC_50_ = 85.91 ± 0.24 µg/mL followed by the SPDCM fractions with IC_50_ = 182.50 ± 0.35 µg/mL, and n-Hexane fractions were reported to be the least active between the tested samples in comparison to ascorbic acid of IC_50_ = 67.14 ± 0.25 µg/mL for DPPH and IC_50_ of 69.96 ± 0.18 µg/mL for ABTS assay. In the in vivo activities, the SPM extract was the most effective with 55.14% inhibition as compared to diclofenac sodium with 70.58% inhibition against animals. The same SPM crude extract with 50.88% inhibition had the most analgesic efficacy as compared to aspirin having 62.19% inhibition. Hence, it was assumed from our results that all the tested samples, especially the SPM and SPDCM extracts, have significant capabilities for the investigated activities that could be due to the presence of the bioactive compounds. Further research is needed to isolate the responsible chemical constituents to produce innovative medications.

## 1. Introduction

Medicinal plants are used throughout the globe for their multiple health benefits from the emergence of human civilization to date [1]. The therapeutic herbs are capable of producing many bioactive compounds attributed with promising antimicrobial, free radical scavenging, anti-inflammatory, analgesic, and numerous other healing properties against a variety of diseases [2]. The therapeutic significance attributed to the medicinal herbs is due to the presence of flavonoids, phenols, alkaloids, and other bioactive chemical ingredients [3,4,5]. These natural products help the human body in performing normal physiological functions [6]. The flavonoids and phenols serve as a potent source to scavenge the free radicals, resist microbes, cure inflammation and allay pain [7]. Alkaloids are known to perform metabolic roles. Hence, they mainly function in animal physiology [8]. Besides this, they also function as steroidal drugs, which are known as steroidal alkaloids [9]. The genus *Scutellaria* is mainly represented by flavonoids and phenolic constituents [10], such as glucuronides comprises baicalin and wogonin-7-Oglucuronide or aglycones that are wogonin-7-Oglucuronide baicalein and wogonin. Phenolic constituents can scavenge the free radicals, resist the pathogenic microbes and inhibit the platelet aggregation [11,12]. Saponin works as an expectorant, emulsifying and antifungal agent [13,14]. The leading role of steroids is to stimulate sex hormones [15]. Only one-third of infectious diseases are treated with the help of synthetic products [16]. In comparison to chemical drugs, natural drugs are reported to produce excellent and effective results having very less or no side effects. The emergence of multidrug-resistant microbes has limited the availability and affordability of numerous recommended marketed antibiotics over the globe [17]. Therefore, it diminishes the effectiveness of the medication procedures and increases the rate of morbidity, mortality, and enhances human health care expenses [18]. To overcome, the antimicrobial complications the researchers continue their efforts to search for new sources from the plants which is eventually a significant basis for the production of modern medication to overwhelm the socioeconomic and human health effects instigated by the microbes and other oxidative stress [19]. Multiple investigations emphasize that several natural plant-based antioxidant agents have been useful in addressing health complications associated with oxidative stress [20]. Some plants basis antioxidant sources have demonstrated promising biological impacts; having the capability to resist the human pathogenic microbes, relieve pain, and heal inflammation [21].The development of an effective anti-inflammatory medication product with a higher margin of safety has always been a challenge [22]. The pathogenic complications also lead to inflammation which is a recent and most terrifying challenge for the scientist to explore innovative products to overcome inflammations [23] for the reason that the intake of synthetic medications for the long term may produce adverse effects [24]. Thus, it is the basic need to search for an effective natural remedy that has the potential to cure inflammation and relieve pain with less side effects [25]. Despite recent advancements in pain medicines, scientists continue to seek out safe, effective, and strong analgesic medications derived from plants since they are known to produce low side effects [26,27]. The genus *Scutellaria* L. (Lamiaceae), also known as skullcap, consist of around three hundred and fifty plant species and is practiced as a traditional remedy in several local communities. The genus *Scutellaria* is cosmopolitan by habitat and is mainly found on the tail of mountains in the mildly hot and humid areas of East Asia, America, and Europe [28]. In Pakistan the *Scutellaria* species are mostly distributed in Swat, Chitral, Mansehra, and Parachinar [2,29,30,31]. The plant species of the genus *Scutellaria* used as traditional remedies and also depicted promising capabilities to resist microbes, purify the blood, regulate the menstrual cycle, and alleviate inflammation and relieve pain [32]. The mentioned feature of the genus is attributed due to the presence of a diverse range of responsible chemical ingredients: scutellarin, alkaloids, baicalin, tannins, saponins, and glycosides, which are well known for multiple health benefits, including antimicrobial and antioxidant, as well as for their substantial capabilities for the treatment of inflammation and pain [2]. The antimicrobial resistance of the *Scutellaria* species has been of great importance among investigators. The search for combinations of new anti-inflammatory and analgesics among vast sources of medicinal plants is intensifying. The main reason is that this type of data ensures the discovery of therapeutic medicines that can be recently produced and have the ability to scavenge the free radicals, cure inflammation and suppress, reduce, or alleviate pain [33,34]. *Scutellaria petiolata* (Lamiaceae) is a perennial chasmophytic, suffruticose herb with a hard woody rootstock and slender, erect, round–quadrangular simple or much-branched, eglandular leafy stem. The leaves are petiolate, along with thick-textured inferior and cauline ovate or generally ovate, entire, crenate or serrate, usually cuneate or particularly cordate acute to obtuse; the upper bract as an elliptic is full. Inflorescence is lax or more reduced, apical or lateral. Scattered flowers are in the axil of the bract, similar to leaves. The calyx is five, lacks the scutellum with expanding fruit, and the corolla are also five in number, having violet-blue color, with upright, pilose, and sessile glands. The nutlets are finely tuberculate together with a small fruit tuft of mostly long propagating multicellular eglandular hairs, mostly black. The flowering duration range from mid-May to September [35]. *Scutellaria petiolata* mainly distributed in colder regions on mountain tails of India, Afghanistan, Kashmir, and Pakistan. In Pakistan, *S. petiolata* is frequently found on the mountain tails of District Swat and Chitral [35].

The dried powder of the selected plant (root, stem, leaves, and flowers) is used by the local communities to overcome antimicrobial disorders, cure inflammation, and a remedy to relieve pain but still needs precise scientific corroboration.

Hence, the current investigation is carried out for the first time to seek out information regarding the efficiency of the selected plants for various health problems and reintroduce the importance of natural products in various health complications: in general, to capture the attention of scientists by screening them for their phytochemical composition, in vitro, antimicrobial, antioxidant capabilities, and in vivo pharmacological; anti-inflammatory and analgesic significance for *S. petiolata* for which they have never been examined earlier; and update the literature of genus *Scutellaria*. 

## 2. Materials and Methods

### 2.1. Apparatus and Reagents Used

Methanol, dichloromethane, n-Hexane, and dimethyl sulfoxide (Fisher Scientific, Loughborough, UK), distilled water (Milli-Q, 31PB, France), rotavapor (BUCHI, 2017, Switzerland). ESI-LC-MS/MS (LTQ XL, Thermo Electron Corporation, Waltham, MA, USA) and Xcalibur 2.2 software (Thermo Fisher Scientific, Waltham, MA, USA) were purchased from Fisher Scientific (Illkirch, France). The solvents used for liquid chromatography were ESI-LC-MS grade acetonitrile (Fisher Scientific). 

### 2.2. Collection and Identification of Plant Samples

The selected plant *Scutellaria petiolata* specimens were gathered (May–July 2018) from various spots of Kalam, District Swat, Khyber Pakhtunkhwa, Pakistan. The specimens were identified by the taxonomist Dr. Jan Alam Associate Professor, Department of Botany, Hazara University KP, Pakistan using available literature [35]. The plant specimen was properly preserved and placed in the herbarium Department of Botany (Herb⁄ HU⁄6642) Hazara University Mansehra Pakistan for future studies.

### 2.3. Plant Samples Processing

The *Scutellaria petiolata* specimens were cleaned with tap water to eradicate the useless materials and then placed under shade for complete dryness at room temperature to avoid the loss of essential and volatile ingredients. 

#### 2.3.1. Crude Extract Preparation

The dried plant samples were shredded with an electric grinder and gained 3700 g of fine powder. The powder was kept in covered bags in the refrigerator, at the temperature of 4 °C until needed. A lot (2700 g) of plant powder were soaked with 6 L, commercial-grade methanol (MeOH) in glass containers for crude extract preparation. The glass container was consistently agitated for 21 days and then the mixture (solubilize plant material and MeOH) was poured into a conical flask through Whatman filter paper. The remaining material on the filter paper was once again submerged in a mixture of 10%, water, and MeOH for an additional 21 days and repeated a similar process to get hold of the filtrate. The filtrates taken from both processes were mixed and evaporated at a temperature of 40 °C and 120 rpm via a rotary evaporator for the evaporation of the solvent MeOH. Eventually, crude extract in the form of semi-solid paste was obtained containing solvent which was further administrated in the hot water bath at 40 °C to remove the entire MeOH solvent. The same procedure was adopted for filtrates obtained again, and in the end, the crude extract (SPM) of 640 g proceeded further for fractionation.

#### 2.3.2. Fractionation of Crude Extract

The crude extract of 600 g was homogenized with 1 L of distilled water and then shaken with the same quantity (1L) of n-Hexane (SPNH) and dichloromethane (SPDCM) using a separating funnel through solvent-solvent extraction. The formation of the clear band was observed among the solvents formed and each part was collected separately and passed through a rota evaporator using the same temperature of 40 °C and 120 rpm to obtain the targeted fractions. Eventually, the SPNH and SPDCM dry mass fractions at a quantity of 23 g and 31 g were obtained respectively. The SPAQ was noticed and yielded in a maximum quantity of 39 g as compared to the fractions.

### 2.4. Stock Solution

The crude extract (SPM) and each fraction (SPNH, SPDCM, and SPAQ) of *S. petiolata* at a mass of 2 g were homogenized in 10 mL DMSO (99.99%) to arrange a stock solution for the various phytochemicals and biological activities and placed in refrigerator till further use.

### 2.5. Qualitative Assessment of Phytochemical Detection

The *S. petiolata* extracts SPM, SPDCM, SPNH, and SPAQ were proceeded for the detection of various phytochemical analyses to validate the presence of representative groups of flavonoids, alkaloids, phenols, and carbohydrates applying the standard techniques [2,36,37]. 

#### 2.5.1. Flavonoids

The presence of the flavonoids group was detected by adding a few drops in SPM, SPDCM, SPNH, and SPAQ of *S. petiolata* with 5% of NaOH solution, and then, a few drops of the HCl were added. Consequently, the changes from yellow to colorless showed the presence of flavonoids.

#### 2.5.2. Phenols

A few drops from the stock solution of extracts SPM, SPDCM, SPNH, and SPAQ were treated with FeCl_3_ solution in glass tubes and then shaken. The change of the mixture to bluish-green color indicates the presence of the phenols group.

#### 2.5.3. Alkaloids

The presence of alkaloid groups in the extracts SPM, SPDCM, SPNH, and SPAQ was carried out by the addition of 0.5 mL from the stock solution with 2% of the H_2_SO_4_ solution. The mixture was then placed in a hot water bath and after 3 min few droplets of Dragendorff’s reagent were added, and the glass tubes were placed aside for coolness. The change of mixture into orange-red color precipitate indicates alkaloids group.

#### 2.5.4. Carbohydrates

In SPM, SPDCM, SPNH, and SPAQ extracts of *S. petiolata*, the presence of the carbohydrate was determined by the addition of 3 mL of the tested samples stock solution along with 2 mL of research-grade Benedict’s reagent in glass tubes. Later, the obtained mixture was kept in a hot water bath for 3 min. Finally, the color change to reddish-brown precipitates indicates the presence of the carbohydrate in the *S. petiolata* extracts.

### 2.6. Analytical Determination of Flavonoids and Phenols

The *S. petiolata* SPM, SPDCM, SPNH, and SPAQ extracts were profiled for the analytical evaluation of the total flavonoids (TFC) and total phenolic contents (TPC) using available literature standard procedure [2,38].

#### 2.6.1. Quantification of Total Flavonoids Contents

For the flavonoid quantification in the tested samples SPM, SPDCM, SPNH, and SPAQ extracts of *S. petiolata* was represented as mg QE/g, and equivalent to the dry mass of the samples. About 9 mL of the distilled water (DW) was added with 1 mL from the stock solution of tested samples extract stock, with further addition of 1 mL from the 5% of NaNO_2_ in a glass test tube and then placed to incubate for 6 min. Next, around 2 mL from the 10% of AlCl_3_ was mixed into each sample in test tubes and then placed undisturbed for 5 min. Lastly, about 2 mL from 1 M of NaOH was put into each of the tested samples in the test tubes. The absorbance was checked at 510 nm employing a UV-visible spectrophotometer.

#### 2.6.2. Estimation of Total Phenolic Contents

The Folin–Ciocalteu reagent (FCR) assay was used to determine the TPC in various extracts (SPM, SPDCM, SPNH, and SPAQ) of *S. petiolata*. The TPC was calculated as mg GAE/g, of the dry mass of the tested samples. The tested samples of 5 mg were added with 5 mL distilled methanol and then mixed with 10 mL of DW and kept undisturbed for 5 min. Around, 1 mL from the tested sample was taken in a glass test tube with the addition of DW till the volume of the samples reached 10 mL. After that, 1 mL of FCR was added up to each sample and incubated for 6 min. Right after the incubation, 10 mL from the 7% of Na_2_CO_3_ solution was mixed. Eventually, the final volume of the reaction mixture (tested samples) reached 26 mL with the further addition of DW and placed to incubate for 90 min at room temperature. The sample absorbance was observed at 760 nm via a UV-visible spectrophotometer.

### 2.7. ESI-LC-MS/MS Analysis

The SPM extract of *S. petiolata* was investigated to highlight its chemical ingredients via ESI-LC-MS analysis through linear ion trap mass spectrometer connected with electrospray ionization (ESI) source applying both positive and negative ionization mode through standard method [10,39]. The tentative identification of the compounds was attained using direct injection mode utilizing ESI. The capillary voltage was fixed at 3.3 kV at 280 °C, however, the sample flow rate was kept at 10.5 µL/min. The mass range was adjusted as 50–2000 *m*/*z*. The collision-induced dissociation (CID) energy varied from 10 to 30, while during MS/MS the energy range was maintained at 10–45, based on the nature of the parent molecular ion. In the mobile phase, the MeOH and acetonitrile ratio was fixed at 80:20 (*v*/*v*). Furthermore, the MS parameters for each compound were properly regulated to authenticate the extremely satisfactory ionization and ion transfer conditions. However, the best possible signals of both the precursor and fragment ions were achieved by infusing the analytes and manually modifying the parameters. The parameters sources were the same for all the analytes.

### 2.8. Biological Activities

The studies plant in extracts SPM, SPDCM, SPNH, and SPAQ were screened for their significance against the microbes and examined their potential to act as an antioxidant agent. 

#### 2.8.1. In Vitro Activities

In vitro biological activities including antimicrobial and antioxidants were performed to report the significance of *S. petiolata* extracts. The antimicrobial activity was carried out by using the bacterial strains; *S. typhi*, *K. pneumonia*, *B. atrophaeus*, *B. subtilis*, and fungal strains; *A. parasiticus* and *A. niger*. The microbial strains were taken from the Microbiology Lab and were properly identified by Dr. Hazir Rahman (Microbiologist) Department of Microbiology, AWKUM, Mardan. 

#### 2.8.2. Antibacterial Evaluation

The antibacterial potential of the *S. petiolata* extracts (SPM, SPDCM, SPNH, and SPAQ) was examined by agar well diffusion assay using the reported literature with slight modifications [2,40]. 1 mg from the tested samples were homogenized in 1 mL of Dimethyl sulfoxide (DMSO) to get 1000 ppm solution from where 50 and 100 µL volume were used for further analysis. To proceed with the antibacterial assessment nutrient agar media was prepared by adding 28 g of the nutrient agar with 1 L of distilled water followed by vigorously shaking until the media completely dissolved. The NA media, wire loop, well borer, and glass Petri dishes were autoclaved at 121 ℃ for 15 min for sterilization. Right after sterilization, the nutrient agar media of around 20 mL agar was poured into each Petri plate in the laminar flow hood until it solidifies. The bacterial strains (*S. typhi*, *K. pneumonia*, *B. atrophaeus*, *B. subtilis*) were properly inoculated via a wire loop using a safety kit keeping the concentration of bacterial cell density of (1.5 × 108 CFU/mL). On the solidified NA media of each Petri plate, five wells of equal size of 3 mm were made using a cork borer at the same distances from each other. The tested samples SPM, SPDCM, SPNH, and SPAQ at dosages of 50 µg⁄ml and 100 µg⁄ml was poured into the well first and second respectively while the third and fourth well was filled with the standard (Levofloxacin & Erythromycin) of equal concentration 50 µg⁄ml and 100 µg⁄ml) for the Gram-positive and Gram-negative strains, correspondingly. The DMSO which was used as a negative control was poured into the fifth well on the agar media. All the glass Petri dishes were kept overnight in the incubator at a temperature of 37 °C. Finally, the Petri plates were taken out from the incubator, and the zone of inhibition (ZOI) around the well was measured in mm. The entire data were taken in triplicates, statistically analyzed, and represented as mean ± SEM.

#### 2.8.3. Antifungal Assessment

The antifungal potential of the tested samples of *S. petiolata* was conducted using an agar well diffusion assay as described by Shah et al. [2]. Dosages from the SPM, SPDCM, SPNH, and SPAQ extracts at concentrations (50 µg/mL and 100 µg/mL), as earlier mentioned for antibacterial activity. Thirty-nine (39) grams of potato dextrose agar (PDA) was added with1 L of distilled water in a conical flask and covered followed by continuous shaking till it homogenized. All the required materials comprising PDA media, glass Petri plates, wire loop, and steel borer (3 mm) were carefully autoclaved at 121 °C for 20 min. After sterilization, the media of around 20 mL was poured into each Petri plate under aseptic and then kept undisturbed till to solidify the media. The fungal inoculant (*A. parasiticus* and *A. niger*) at a concentration of 108–109 CFU/mL was spread over the solidified PDA media and five holes of equal size (3mm) were made at the same distance from each other. The extract dosage of 50 µg/mL and 100 µg/mL was injected into wells 1 and 2, and the antifungal standard (Fluconazole) at the same concentration 50 µg/mL and 100 µg/mL was added to the third and fourth wells, and the fifth hole was filled with the negative control (DMSO), respectively. After that, the Petri dishes were properly packed and incubated for 72 h at 25 °C in the incubator. Finally, the Petri plates were taken out and ZOI was calculated around the hole in mm. All the antimicrobial data were taken in triplicates and represented as the mean ± SEM.

#### 2.8.4. Antioxidant Determination

The free radicals scavenging significance in the tested fractions (SPM, SPDCM, SPNH, and SPAQ) were examined using the most reported 2,2-diphenyl-1-picrylhydrazyl (DPPH) and 2,2 -azino-bis (3-ethylbenzothiazoline-6-sulfonic acid (ABTS) bioassay [2,41,42]. The antioxidant capacity was performed through a DPPH assay by taking 3 mg of DPPH and homogenizing properly 100 mL distilled methanol and placing an undisturbed to form the free radicals in the solution in dark for 30 min. The concentrations (1000, 500, 250, 125 and 62.5 µg/mL) of the tested samples including standard ascorbic acid were properly prepared. Later, 2 mL from the SPM, SPDCM, SPNH, and SPAQ extracts, and ascorbic acid were added to the 2 mL stock solution of DPPH and then placed to incubate for 20 min in the dark. Right after the incubation, the absorbance of tested samples was observed at 517 nm using UV/Vis spectrophotometer. The scavenging capability in the tested samples was calculated using equation.
% Free radicals scavenging activity = A − B ⁄A × 100(1)
whereas A represents absorbance of the control and B represents the absorbance of the standard.

ABTS assay was also used to analyze the free radical scavenging potential of the tested samples. To proceed with the ABTS assay, around 383 mg of ABTS and 66.2 mg of the K_2_S_2_O_8_ were separately homogenized in 100 mL of the MeOH, and then, both were mixed. After that, 2 mL of the mixture were placed to incubate with 2 mL of tested samples at a concentration of (1000, 500, 250, 125, and 62.5 µg/mL) for 25 min. Finally, the absorbance of the tested samples was measured at 746 nm via UV spectrophotometer, and the results were estimated using Equation (1).

### 2.9. In Vivo Activities

The selected plant was screened in various tested samples to examine their in vivo anti-inflammatory and analgesic significance.

#### 2.9.1. Ethical Approval

Ethical approval for the experimental animals (Swiss albino mice) with body weight (B.w) around (25–30 g) was taken from the official ethical committee of Abdul Wali Khan University Mardan (AWKUM) with (Ref. No: AWKUM/Bot/2018/1679, Dated: 8 November 2018), Department of Botany, AWKUM (if needed will be provided). The experimental animals were purchased from the veterinary research institute Peshawar and kept under an aseptic condition in rubber cages for around 45 days at a maintained temperature of 20 °C at the animal house of AWKUM following ARRIVE guidelines as stated by Du et al. [43] 

#### 2.9.2. Anti-Inflammatory Activities

The selected plant in various extracts (SPM, SPDCM, SPNH, and SPAQ) was examined for anti-inflammatory activities using the standard method with slight modification [2]. To progress the anti-inflammatory activity, 30 mice in 5 groups were taken with 6 experimental animals in each group for the tested samples (plant fractions, negative control, and standard).

To induce paw edema, carrageenan 1 mL was injected into the paws of all six groups of the Swiss albino mice. Right after 30 min, around 1 mL of the normal saline (NS) was infused into the paw of the group second mice to estimate the anti-inflammatory efficacy, and then 1 mL of the standard diclofenac sodium (DS) was injected to the paw of third experimental animals’ group. Furthermore, the dosage at concentrations of 50 and 100 mg/kg/body weight (BW) was injected into the 4 and 5 groups of the experimental animals correspondingly. The paw diameter of the Swiss albino mice was calculated and observed after 1, 2, and 3 h, respectively, and represented% inhibition using Equation (2).
% Inhibition of the tested samples = a− b ⁄a × 100(2)
where (a) inducer (carrageenan) and (b) inhibition of the tested sample (crude oils, standard, and control), whereas in analgesic activity (a) represents the writhes inducer acetic acid.

#### 2.9.3. Analgesic Activities

The analgesic capacity of *S. petiolata* in extracts (SPM, SPDCM, SPNH, and SPAQ) was examined using acetic acid-induced writhing bioassay using experimental animals (Swiss albino mice) [2,41]. To proceed with the analgesic activity, 30 mice were taken and evenly arrange into five groups and each group contain 6 mice. The tested samples SPM, SPDCM, SPNH, and SPAQ along with the negative control, and standard were injected into the Swiss albino mice through intraperitoneal muscle via an authorized size sterilized syringe. Around 1 mL of acetic acid was infused into each group (1–5) of the Swiss albino mice. Right after 30 min, 1 mL of NS (negative control) and 1 mL of the standards (aspirin) were injected into the second and third groups of the experimental animals respectively. Furthermore, the fractions SPM, SPDCM, SPNH, and SPAQ at dosages 50 and 100 mg/kg B.W were injected into the experimental animals of groups 4 and 5 accordingly.

The writhes induced through the acetic acid were calculated and compared with the negative control and standard for 10 min. The results were obtained through Equation (2), authentic through statistical analysis, and expressed as% inhibition.

### 2.10. Statistical Analysis

The recorded data were analyzed through one-way analysis of variance (ANOVA), following Bonferroni’s and the significance level: *p* = 0.05 represented as (*) and 0.01 denoted with (**) using two-way ANOVA, while Tukey’s multiple comparison test, ns = >0.9999, *p* = **** <0.0001 was followed for antimicrobial activities. However, the antioxidants significance was determined through a nonlinear regression graph (NLRG) and was designed among the% inhibition and dosages of the tested samples, and the IC_50_ was estimated via GraphPad Prism 9 software for Windows (GraphPad-Software, San Diego, CA, USA, 2020) through the equation below:Y = 100/1 + (ˆHill Slope)
where 1 denotes the inhibitors, Y for the inhibitor’s reaction, and Hill Slope demonstrates the steepness of the curves.

## 3. Results and Discussion

Medicinal plants are an affluents basis for the mass production of drugs. The medicinal plants are selected based on the information regarding ethnopharmacology collected from the local practitioners and communities [44]. Plant’s traditional uses are authenticated by screening the composition of the bioactive ingredients followed by their in vitro and in vivo studies, which are fundamental procedures for determining a plant’s therapeutic potential [45]. Medicinal plants contain responsible bioactive ingredients, such as flavonoids, phenols, and alkaloids. These chemical constituents have multiple health benefits [46]. Environmental variables have a significant impact on the quantity of phytochemical ingredients [47]. Light, soil fertility, soil water, temperature, and salinity, which are known as the external factors and variables, have a significant impact on some of the processes that are mostly that are linked with the development and growth of the plants. Moreover, the ability to produce different secondary metabolites thus causes alterations in the entire phytochemical profiles that are involved in bioactive substance generation [48]. To put it another way, plants can produce secondary metabolites gradually according to the stress created due to the environment. The secondary metabolism of plants can be thought of as a plant’s ability to adapt and survive in response to environmental stimuli throughout its life [49]. 

### 3.1. Phytochemicals Qualitative Detection 

Phytochemical detection of the representative groups to highlight the significance of *S. petiolata* based on which they proceeded to various biological studies. Flavonoids, phenols, alkaloids, and carbohydrates were found in all tested samples except in the SPNH fraction in which flavonoids, alkaloids, and carbohydrates were not detected. Additionally, from the examinations, it has been found that the flavonoids were found in significant amounts in all extracts rather than the other bioactive compounds. From the results, the aqueous extract presents negligible or lower positive results, whereas a moderate amount of bioactive was present in the other fractions. The current findings support the literature stated by Shah et al. [2] for *S. edelbergii*. Flavonoids are the dominant group in the genus *Scutellaria* as reflected in the data noticed by Shang et al. [32]. The findings are inconsistent with the data published by Stepanova et al. [50] for phenols who identified the most active bioactive compounds in *Scutellaria*. The results showed that among all the identified bioactive compounds, flavonoids are largely present. The results are clarified with the help of data presented in Table 1.

### 3.2. Assessment of Total Flavonoids and Phenols

The understudy plant in extracts SPM, SPDCM, SPNH, and SPAQ was investigated for the quantitative estimation of total flavonoids and flavonoid contents.

#### Flavonoids and Phenols Contents

The determination of *S. petiolata* in various extracts revealed that a significant quantity of flavonoids was around 78.2 ± 0.22 ** mg QE/mg in the SPM extract, followed by the SPDCM and SPAQ extracts with a quantity of 67.4 ± 0.15 ** mg QE/mg, and 49.4 ± 0.37 ** mg QE/mg, respectively, while the least quantity of flavonoids was observed in the SPNH fraction extract, which was found to be about 46.2 ± 0.66 ** mg QE/mg. Similarly, the SPM extract also consists of significant levels of total phenols of 66.2 ± 0.33 ** mg GAE/g, followed by SPDCM and aqueous extract with the quantity of 61.5 ± 0.66 ** mg GAE/g, 41.3 ± 0.33 ** mg GAE/g for phenols, respectively, shown in Table 2. In addition to that, the minimum quantity of phenols was recorded in the SPNH fraction of *S. petiolata.* According to Table 2, the SPNH fraction displayed negligible levels in both flavonoids and phenols in comparison to the other extract. The current outcomes are correlated with that of the research conducted by the researchers Liu et al. [51] and Shah et al. [2] for *Scutellaria platystegia* and *S. edelbergii* respectively to determine their phenolic and flavonoid contents. The current findings depicted a little variation with the screening described by Park et al. [52] and Chen et al. [53] for *S. baicalensis*. The variation in the quantity of the total phenols and flavonoids is mainly influenced by edaphic, climatic, habitat, quantity, and quality of water available to the plants as stated by Li et al. [54]. The active biological compounds that exist within the medicinal plant extract that was chosen for the research and that show positive outcomes for the constituents analyzed can be best estimated with the help of phytochemical analysis. These possess a medicinal role, which is essential in maintaining the stability of human physiology. The extracts of *S. petiolata* in crude extract and subfractions with a notable content of phytochemicals. It is highly effective in many biological activities and remedial signs [13]. The plants that were chosen to consist of an average amount of phenol, alkaloid, and flavonoids. These are excellent sources for different diseases and are essential for normal physiological activities of the human body, a finding agreed with Ghosh et al. [55].

### 3.3. ESI-LC-MS Assessment

Based on the most substantial abilities for the observed biological activities the most active SPM extract was profiled to identify the promising bioactive ingredients tentatively. The tested sample contains 19 compounds signifying four groups: flavonoids (**10**), alkaloids (**3**), terpenoids (**5**), and phenols (**1**), among which, 9 compounds were detected in negative ionization mode (NIM) among, and 10 bioactive compounds were observed in positive ionization mode (PIM). The Full Scan chromatograms of negative ionization mode and positive ionization mode and chromatograms of the respective compounds are given in Appendix A. 

Flavonoids were noted as a major group and contributed ten (**10**) compounds among which six (6) compounds (**2, 4, 6, 7, 8,** and **12**) were identified in NIM, while four compounds (**15**, **16, 17,** and **18**) were observed in PIM as displayed in Table 3 and Figure 1. These compounds were reported for the first time from *S. petiolata* while earlier reported from *S. altissima*, *S. baicalensis*, *S. barbata* [56,57,58], *S. baicalensis* [59] *S.multicaulis* [60] *S. prostrata* [32,61] *S. barbata* [62] *S. multicaulis* and *S. patonii* [60] and possess the promising potential to resist the human pathogenic microbes, scavenge the free radicals treat inflammation [63,64,65]. 

In addition to that five terpenoids (**9, 10, 11, 13,** and **14**) were observed in PIM in the most active extract previously reported from *S. drummondii*, *S. rubicunda*, and *S. barbata* [62,66,67] having the therapeutic significance to act as an antioxidant, antibacterial, and anti-inflammatory agent [32,68,69]. The SPM contains three alkaloids two compounds (**1** and **3**) were noticed in NIM and one compound (**5**) was observed in PIM previously reported from *S. flavescens* [70] and is well-known for its antioxidant, antimicrobial, anti-allergenic, analgesic, and anti-inflammatory [71,72,73], while one compound **19** observed in PIM was representing the phenolic group which was previously reported from *S. baicalensis* [74] and can inhibit microbial growth [75]. 

### 3.4. Antimicrobial Potential

#### 3.4.1. Antibacterial Significance

The local practices were validated due to the current antibacterial assessment utilizing *S. petiolata* in various fractions as represented in Figure 2. The highest resistance against the Gram-negative bacterial strain *S. typhi* was demonstrated by the SPM extract with 17.8 ± 0.04 mm zone of inhibition (ZOI), proceeded with 17.2 ± 0.02 mm and 16.8 ± 0.04 mm ZOI by the SPDCM and SPNH fractions respectively, while the least resistance was offered by the aqueous extract with 16.4 ± 0.03 mm from low to high dose. Furthermore, the SPM, SPDCM, SPNH, and SPAQ extract presented 18.8 ± 0.04, 17.3 ± 0.02, 16.6 ± 0.01, and 14.3 ± 0.03 mm ZOI against the *K. pneumonia* correspondingly as compared to levofloxacin which exhibited 23.6 ± 0.02 and 21.3 ± 0.04 mm ZOI against the *S. typhi* and *K. pneumonia* correspondingly. In addition to that, the same SPM extract presented 19.4 ± 0.01 mm ZOI, followed by the SPDCM and SPNH fraction with 18.7 ± 0.02 and 17.6 ± 0.01 mm ZOI against the Gram-positive bacterial strains *B. atrophaeus* while the aqueous extract revealed the least inhibition of 16.2 ± 0.04 mm ZOI in comparison to erythromycin with 21.8 ± 0.03 mm ZOI. The antibacterial significance of the selected plant might be due to the presence of flavonoids as stated by Cushnie et al. [78] and Gorniak et al. [79]. The genus *Scutellaria* also contains phenols that have antibacterial capacities as documented by Cueva et al. [80]. Moreover, the tested samples SPM, SPDCM, SPNH, and SPAQ offered 18.8 ± 0.04, 18.2 ± 0.02, 17.4 ± 0.04, and 16.6 ± 0.02 mm ZOI against *B. subtilis* as compared to standard having 23.2 ± 0.05 mm ZOI. The recent studies uncovered that significant inhibition was exhibited against the Gram-positive bacterial strains as compared to the Gram-negative bacterial strains consented with the findings of Shah et al. [2] which screened *S. edelbergii* and also the literature stated by Leach et al. [81] for *S. baicalensis.* Our recent data was not equated to the findings of Arituluk et al. [82], Yilmaz et al. [26], and Ordan et al. [83] for some *Scutellaria* species. 

#### 3.4.2. Antifungal Significance

The antifungal significance of the selected plant in various fractions was tested and reflected appreciable outcomes as displayed in Figure 3. The SPDCM extract presented the maximum resistance of 19.07 ± 0.02 mm, followed by the SPM and SPNH fractions of 19.07 ± 0.02 mm and 16.03 ± 0.01 mm ZOI respectively against the *A. parasiticus* while the least inhibition among the tested samples were offered by the aqueous extract with 15.11 ± 0.04 mm in comparison to fluconazole with 20.8 ± 0.05 mm ZOI. Furthermore, the SPDCM, extract exhibited 18.87 ± 0.04 mm ZOI proceeded by the SPM, SPNH, and SPAQ extract with 18.21 ± 0.03, 15.81 ± 0.01 and 14.51 ± 0.01 mm, respectively, in comparison to the standard with 21.5 ± 0.02 mm zone of inhibition. The antifungal activity of the selected plant in various fractions might be due to the presence of an abundant quantity of flavonoids and phenols as stated by Simonetti et al. [84] and Galeotti et al. [85]. Moreover, our recent data are not similar to the literature stated by Shah et al. [2] and Da et al. [86] might be due to the presence of less amount of the secondary metabolites that are reported for the antifungal significance. Our study consented to the findings of Kasaian et al. [69] and Zhao et al. [87] for some species of the genus *Scutellaria.* The same genus *Scutellaria* contains similar compounds that can resist fungal growth.

### 3.5. Antioxidant Capacity

Synthetic free radical (2,2′-azino-bis (3-ethylbenzothiazoline-6-sulfonic acid)) (ABTS) and DPPH tests were used to determine the antioxidant properties of the *S. petiolata* extract and fraction (Figure 4 and Figure 5). In DPPH, the MeOH extract had the maximum antioxidant capacity, with IC_50_ = 78.75 ± 0.19 µg/mL followed by the DCM, n-Hexane, and aqueous fraction which offered an IC_50_ = 140.50 ± 0.20, 192.70 ± 0.15, and 283.10 ± 0.24 µg/mL free radicals scavenging capacities, respectively. In addition to that, in the ABTS assay, the same MeOH extract presented an IC_50_ = 85.91 ± 0.24 µg/mL proceeded by the DCM, n-Hexane, and aqueous fractions having IC_50_ = 182.50 ± 0.35, 224.50 ± 0.13, and 317.40 ± 0.26 µg/mL. It was observed that the n-Hexane fractions were found to be the least active in both ABTS and DPPH assays.

The standard utilized ascorbic acid, which had IC_50_ values of 67.14 ± 0.25 and 69.96 ± 0.18 µg/mL in the DPPH and ABTS assays, correspondingly. The capability to neutralize the free radicals is mainly attributed to the flavonoids and phenolic contents as stated by Yakoub et al. [88] which are the dominant group of bioactive ingredients of the genus *Scutellaria* Malikov et al. [89] and Pei et al. [90]. Several bioactive compounds mainly phenols and flavonoids were extracted from *Citrus limon* L having the capacity to act as antioxidant agents as documented by Imeneo et al. [91] 

The intake of juicy food supplements has the promising capabilities to neutralize the free radicals as stated by Giuffre et al. [92]. The plants with high contents of phenols and flavonoids also offered substantial antioxidant potential as described by Kodama et al. [93]. The current studies reveal that our findings matched with the data stated by Shah et al. [2] and also consented to the data reported earlier by Bazzaz et al. [94] for *S. litwinowii.* Furthermore, it was reflected in previous studies stated by Yang et al. [47] that the plant from the same genus varies in the contents of the responsible bioactive compounds. Thus the outcomes confirmed by Georgieva et al. [95] and Saboura et al. [96] from some *Scutellaria* plant species showed a slight variation as compared to our findings as the edaphic, climatic, and environmental factors alter the composition of the chemical ingredients in plants. 

### 3.6. In Vivo Activities

The selected plant in various extracts was tested to screen their capacity for the treatment to cure inflammation and analgesic significance. 

#### 3.6.1. Anti-Inflammatory Significance

The anti-inflammatory effect of *S. petiolata* in various extracts is presented in Table 4. The carrageenan-induced assay was used to test the anti-inflammatory properties of extracts in Swiss albino mice. Compared to the other extracts that were examined, the SPM fraction was potent 55.14% inhibition cure inflammation, followed by the SPDCM and SPNH with 53.67% and 40.44% inhibitory potential correspondingly. The aqueous fraction, on the other hand, had the least activity 38.97% inhibition. However, conventional diclofenac sodium presented 70.58% inhibition in the experimental used animals is displayed in Table 4. The anti-inflammatory significance of the understudy sample was attributed to the presence of phenolic and flavonoid constituents as stated by Lu et al. [97] and Calvin et al. [98]. Our findings are consistent with those of Lee et al. [99], Liu et al. [51], and Shah et al. [2], who used a similar approach and fractions to explore *S. baicalensis*, *S. barbata*, and *S. edelbergii* respectively. All these plants belong to the same genus and mainly the plant species of the same genus probably contain the same responsible constituents. Our data are inconsistent with the information reported for some *Scutellaria* species, as stated by Varmuzova et al. [100] and Han et al. [101]. The variation among the anti-inflammatory might be due to the approaches used and constituents variation among the plant’s species.

#### 3.6.2. Analgesic Potential

Table 5 indicates the analgesic effect of *S. petiolata* extract and sub-fractions in Swiss albino mice. With 50.88% inhibition, the SPM fraction had the most analgesic efficacy, followed by SPDCM and SPNH fractions with 46.64% and 42.04% inhibition respectively. The aqueous fraction, on the other hand, presented the least with 38.86% inhibition as shown in Table 5. Aspirin was utilized as a standard, and it inhibited writhes generated by acetic acid by 62.19% inhibition. Furthermore, the previous studies stated by Malikov et al. [89] reflected the genus *Scutellaria* plants species contain phenols and flavonoids in maximum quantity. Our findings are consistent with those of Shah et al. [2] for *S. edelbergii*, Lee et al. [99] and Yimam et al. [102] who used similar methods and plant species *S. baicalensis* of the same genus *Scutellaria.* However, our current data does not support the results presented by Uritu et al. [33] and Delazar et al. [103], who screened some plants of the family Lamiaceae to find out their analgesic capacities. The variation among the outcomes might be due to the use of an increased dose of plant extracts, genus and variations among the environmental gradients that alter the composition of the secondary metabolites, as stated by Borges et al. [104].

## 4. Conclusions

It is concluded that *S. petiolata* contains responsible bioactive ingredients with a variety of phytochemicals with a lot of biological properties that might be responsible for its several therapeutic effects. All the tested samples in general and the SPM extract in, have the highest flavonoids and phenolic contents. A total of nineteen bioactive compounds were reported for the first time from *S. petiolata*, among which flavonoids and terpenoids were the dominant groups. Appreciable antimicrobial activity was presented in SPM and SPDCM against the bacterial and fungal strains. The SPM extracts among the screened samples were potent for the DPPH and ABTS free radicals scavenging capacities. The SPM and SPDCMS extracts were observed efficient to cure inflammation and relieve pain in the experimental used Swiss albino mice. Hence, it was concluded that *S. petiolata* might be employed to resist the microbes, scavenge the free radicals, for the treatment of inflammation and pain (analgesic.) These properties are attributed due to the presence of flavonoids and triterpenoids. Still, further investigations are suggested/recommended to screen and isolate the potential chemical ingredients for the examined complications.

## Figures and Tables

**Figure 1 antioxidants-11-01446-f001:**
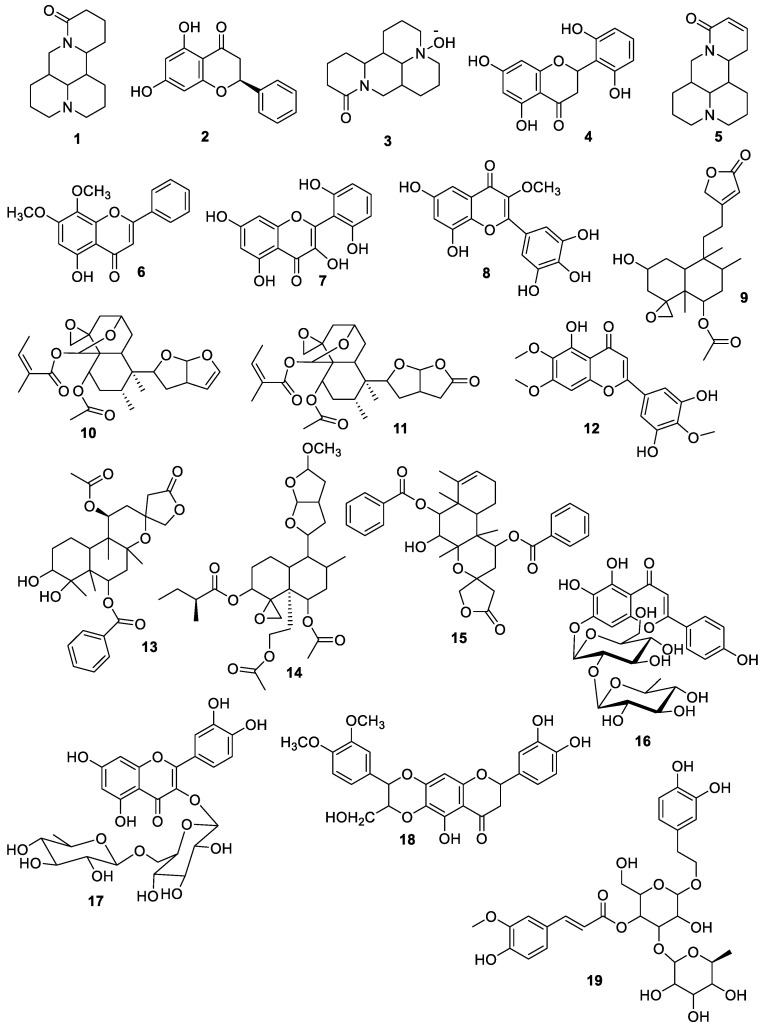
Structures of the tentatively identified compounds in SPM extract of *S. petiolata*.

**Figure 2 antioxidants-11-01446-f002:**
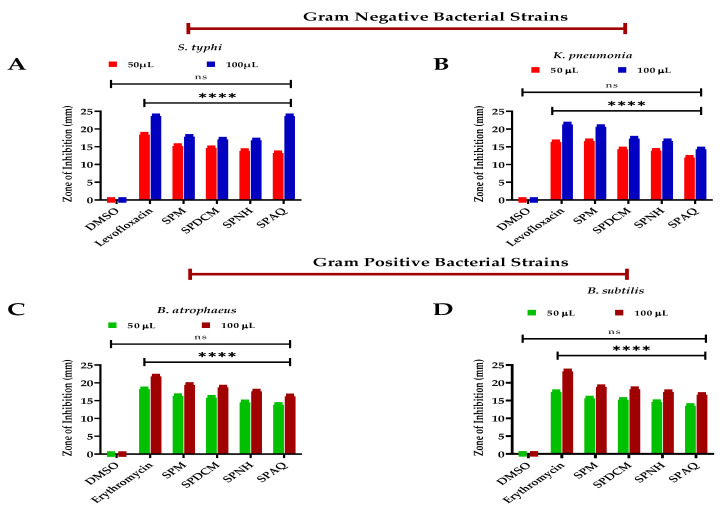
Antibacterial activity of *S. petiolata* extracts, SP) *S. petiolata*, M = methanol extract, DCM = dichloromethane fraction, NH = n-Hexane fraction, AQ = aqueous fraction, DMSO = dimethyl sulfoxide, negative control and positive control = erythromycin and levofloxacin whereas, (**A**) antibacterial resistance against *S. typhi*, (**B**) *K. pneumonia*, (**C**) *B. atrophaeus*, (**D**) *B. subtilis.* All the data was taken in triplicate (*n* = 3) and analyzed through two-way ANOVA, via Tukey’s multiple comparison test, ns = >0.9999, *p* = **** <0.0001).

**Figure 3 antioxidants-11-01446-f003:**
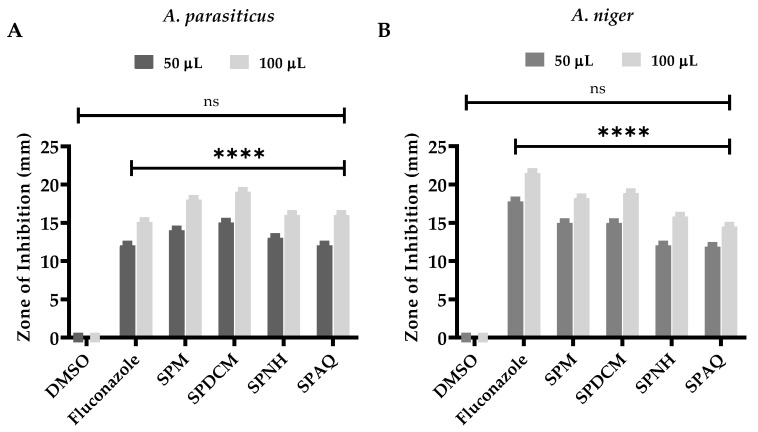
Antifungal activity was of *S. petiolata* crude extract and subfractions, SP = *S. petiolata*, SPM = methanol extract, SPDCM = dichloromethane fraction, SPNH = n-Hexane fraction, SPAQ = aqueous fraction, DMSO = dimethyl sulfoxide negative control and positive control fluconazole, However, (**A**) represent the antifungal significance against *A. parasiticus* and (**B**) *A. niger*. All the data was taken in triplicate (*n* = 3) and analyzed through two-way ANOVA, via Tukey’s multiple comparison test, ns = >0.9999, *p* = **** <0.0001.

**Figure 4 antioxidants-11-01446-f004:**
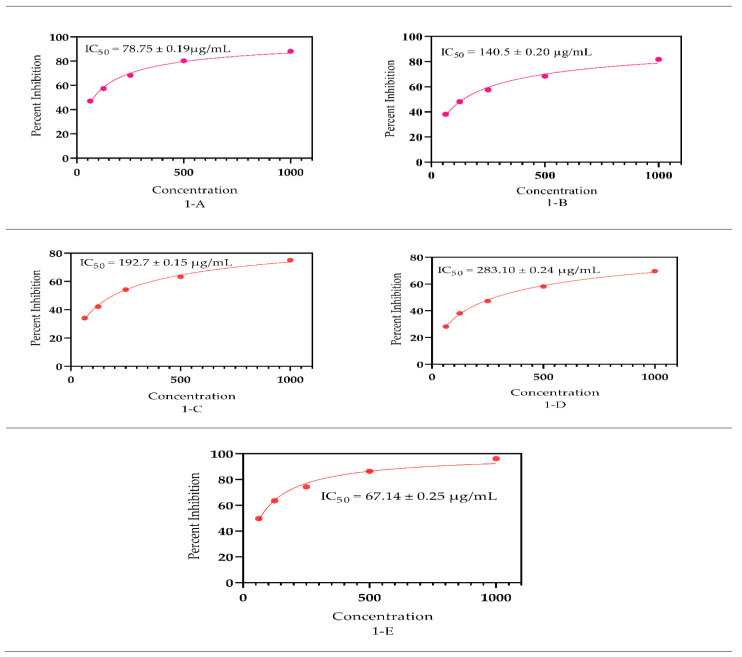
Free radicals scavenging potential by the tested samples of *S. petiolata* using DDPH assay, where **A** represent SPM, **B** SPDCM, **C** SPNH, **D** SPAQ, and **E** standard ascorbic acid significance as an antioxidant agent.

**Figure 5 antioxidants-11-01446-f005:**
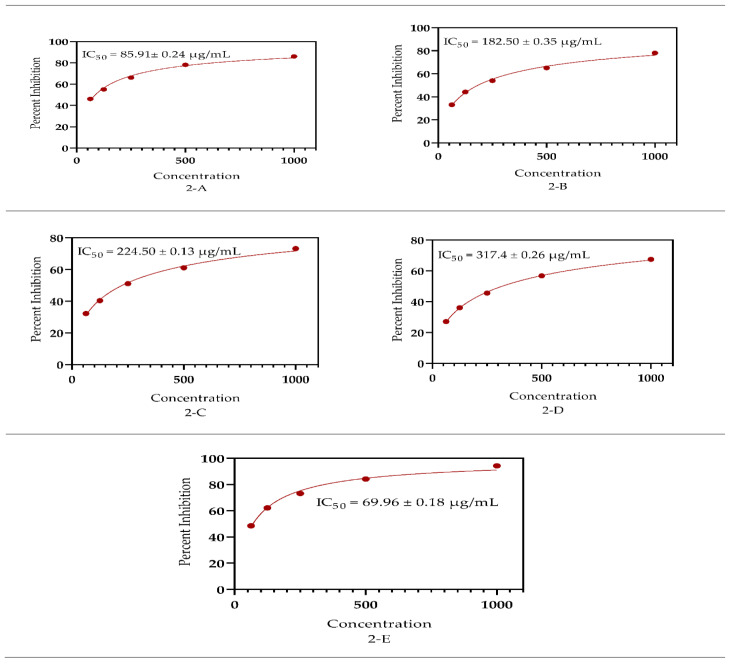
Antioxidant significance by the *S. petiolata* extracts using the ABTS assay, where **A** represent SPM, **B** SPDCM, **C** SPNH, **D** SPAQ, and **E** standard ascorbic acid significance as an antioxidant agent.

**Table 1 antioxidants-11-01446-t001:** Qualitative detection of the phytochemical groups in *S. petiolata* extracts.

Extracts Used	Flavonoids	Phenols	Alkaloids	Carbohydrates
SPM	+	-	+	+
SPDCM	+	+	+	-
SPNH	+	+	-	-
SPAQ	+	+	+	+

SP: *Scutellaria petiolata*, M: methanolic extract, DCM: dichloromethane fraction, NH: n-hexane fraction, and AQ: aqueous fraction, whereas + represents the presence and - shows the absence of the representative phytochemical group.

**Table 2 antioxidants-11-01446-t002:** Determination of the total flavonoids and phenols in *S. petiolata* extracts.

Extracts Used	TFC (mg QE/mg) Dry Fractions Mean ± SEM	TPC (mg GAE/g) Dry Fractions Mean ± SEM
SPM	78.2 ± 0.22 **	66.2 ± 0.33 **
SPDCM	67.4 ± 0.15 **	61.5 ± 0.66 **
SPNH	46.2 ± 0.66 **	38.4 ± 0.33 **
SPAQ	49.4 ± 0.37 **	41.3 ± 0.33 **

SP: *Scutellaria petiolata*, M: methanolic extract, DCM: dichloromethane fraction, NH: n-Hexane fraction, AQ: aqueous fraction, TPC: total phenolic contents, TFC: total flavonoids, QE: quercetin equivalent, GAE: gallic acid equivalent, n: 3, *p* ≤ 0.01 **, mg: milligram, g: gram, the whole data expressed as Mean ± SEM (standard error mean).

**Table 3 antioxidants-11-01446-t003:** Compounds identified through ESI-LC-MS analysis in the active SPM extract.

Numbering	RT (min)	[M−H]^−^/ [M+H]^+^ (*m/z*)	Tentative Identification	Reference Species	Classification
**1**	4.10	247.08	Sophoridine	*S. flavescens* [76]	Alkaloid
**2**	4.22	255.25	Pinocembrin	*S. altissima* [56]	Flavonoid
**3**	4.35	263.08	Oxymatrine	*S. flavescens* [70]	Alkaloid
**4**	4.59	287.17	(tans)-5,7,2′,6′-Tetrahydroxyflavanols	*S. baicalensis* [57]	Flavonoid
**5**	4.62	247.08	Sophocarpine	*S. flavescens* [70]	Alkaloid
**6**	4.71	297.25	5-Hydroxy-7,8-dimethoxyflavone	*S. barbata* [58]	Flavonoid
**7**	4.86	301.08	3,5,7,2,6-Pentahydroxy flavone	*S.baicalensis* [59]	Flavonoid
**8**	5.20	331.25	Myricetin-3’-methyl ether	*S.multicaulis* [60]	Flavonoid
**9**	5.32	393.30	2-Hydroxyajugarin V	*S. drummondii* [66]	Terpenoid
**10**	6.00	489.50	Scutegrossin A	*S. rubicunda* [67]	Terpenoid
**11**	6.10	505.42	Scutecyprol B	*S. rubicunda* [67]	Terpenoid
**12**	6.36	359.20	5,6,2-Trihydroxy-7,8,6-trimethoxy flavone	*S. prostrata* [32]	Flavonoid
**13**	6.62	531.50	Scuterivulactone C2	*S. barbata* [62]	Terpenoid
**14**	7.18	567.50	Lupulin B	*S. linearis* [77]	Terpenoid
**15**	7.55	575.50	Barbatin A	*S. barbata* [62]	Flavonoid
**16**	7.80	595.50	Scutellarein-7-O-neohesperidoside	*S. multicaulis* [60]	Flavonoid
**17**	7.89	611.58	Quercetin-3-O-rutinoside	*S. patonii* [60]	Flavonoid
**18**	8.06	495.33	Scutellaprostin F	*S. prostrata* [61]	Flavonoid
**19**	8.24	639.58	Leucosceptoside A	*S. baicalensis* [74]	Phenol

**Table 4 antioxidants-11-01446-t004:** Anti-inflammatory significance of *S. petiolata* extracts.

Treatments	Change in Paw Diameter (Mean ± SEM)					
	Dose Conc.	after 1 h	after 2 h	after 3 h	Aver. Reading	% Inhibition
Carrag.	1 mL	1.11± 0.03	1.33 ± 0.01	1.66 ± 0.02	1.36 ± 0.12	
NS	1 mL	1.10 ± 0.02	1.31 ± 0.03	1.65 ± 0.01	1.35± 0.02	-----
DS	50 (mg/kg)	0.50 ± 0.02	0.34 ± 0.05	0.18 ± 0.03	0.34 ± 0.02	70.58
SPM	50	0.91 ± 0.06	0.75 ± 0.03	0.62 ± 0.01	0.76 ± 0.03 *	44.11
	100	0.74 ± 0.03	0.61 ± 0.02	0.49 ± 0.03	0.61 ± 0.02 *	55.14
SPDCM	50	0.89 ± 0.07	0.81 ± 0.01	0.70 ± 0.05	0.80 ± 0.05 *	41.17
	100	0.76 ± 0.04	0.63 ± 0.02	0.51 ± 0.06	0.63 ± 0.04 *	53.67
SPNH	50	0.94 ± 0.02	0.88 ± 0.02	0.82 ± 0.03	0.88 ± 0.02 *	35.29
	100	0.87 ± 0.03	0.80 ± 0.02	0.76 ± 0.01	0.81 ± 0.02 *	40.44
SPAQ	50	0.97 ± 0.02	0.92 ± 0.02	0.87 ± 0.03	0.92 ± 0.05 *	32.35
	100	0.88 ± 0.03	0.84 ± 0.02	0.79 ± 0.01	0.83 ± 0.02 *	38.97

Carrag.: Carrageenan, SP: *S. petiolata*: methanol, NH: n-Hexane, DCM: dichloromethane, AQ: aqueous diclofenac sodium = positive control, *n*: 3 with *p* ≤ 0.05 *, data were taken as mean ± SEM.

**Table 5 antioxidants-11-01446-t005:** Analgesic potential of the *S. petiolata* extracts.

Treatments	Dose Conc.	Writhes No. Mean ± SEM	% Reduction in Writhings
AA	1.5 ml	28.3 ± 0.03	
Aspirin	1 ml	10.7 ± 0.02	62.19
SPM	50 (mg/kg)	19.5 ± 0.07 **	31.09
	100	13.9 ± 0.04 **	50.88
SPDCM	50	20.5 ± 0.02 **	27.56
	100	15.1 ± 0.04 **	46.64
SPNH	50	21.6 ± 0.02 **	23.67
	100	16.4 ± 0.03 **	42.04
SPAQ	50	22.2 ± 0.04 **	21.55
	100	17.3 ± 0.05 **	38.86

AA: Acetic acid, SP: *S. petiolata*, M: methanol, NH = n-hexane, DCM: dicholoromethane, AQ: aqueous, aspirin= positive control, *n*” 3 with *p* ≤ 0.01 **, data were taken as mean ± SEM.

## Data Availability

The data presented in this study are available in article and Appendix A.

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
