# Peer review of "Scutellaria petiolata Hemsl. ex Lace & Prain (Lamiaceae).: A New Insight in Biomedical Therapies"

_antioxidants, 2022, doi:10.3390/antiox11081446_

Round 1

Reviewer 1 Report

The manuscript is interesting, however improvements are necessary.

-          In the title and in the whole manuscript:  To avoid confusion, please use the correct and updated botanical nomenclature, for example according to www.gbif.org, and also report the authorship and (in brackets) the botanical family in the title and at the first mention;

-          Abstract section lines 19, 29 and in the whole manuscript, when you have written the numerical values and the standard deviation, sometime you have included spaces before and after the symbol ± and sometime non. Please, be consistent in the whole manuscript, tables and figures;

-          Abstract section, line 37 and in the whole manuscript, separate the numeric value from the unity of measurement;

-          Introduction section line 57, delete one space after … pain;

-          Introduction section, lines 79-86: the color of these lines is too faint, please, verify;

-          Introduction section lines 94-95, please, re-write this sentence because it is not really comprehensible and verify verbs;

-          Introduction section line 98, after ingredients, insert colons instead of semi-colon;

-          Line 98, decide if write the name of molecules in small or in capital letters and be consistent in the whole manuscript;

-          Line 111, delete one space before Scattered;

-          Lines 112-113 and in the whole manuscript, I think it is not necessary to repeat (5) between brackets, the International Reader knows that five is (5); be consistent in the whole manuscript when you have used the same procedure;

-          Lines 115-116, verify the English form of this sentence, in addition, what is the subject of… prefer to grow?

-          2.1 sub section, n-Hexane, please decide if small or capital letter: n-hexane or n-Hexane. Be consistent with other chemicals written in your manuscript;

-          2.1 sub-section line 132 (distilled water), line 136 (deionized water) and in the whole manuscript They are not the same. Which have you used?

-          2.2 sub-section, verify the spacing before and after the hyphen of May –July;

-          2.3.1 sub-section, line 151, delete one space before… The powder;

-          Line 152 and in the whole manuscript, when you write a temperature, separate the numeric value from the symbol: 4 °C and not 4°C;

-          Line 152, replace Twenty-seven hundred (2700) grams … with …. 2700 g;

-          Line 153: … with 6 L, is enough, correct the whole manuscript;

-          Line 162, delete one space before MeOH;

-          Lines 163-164, delete one space after SPM and before the second bracket;

-          Line 164, why 640 grams between brackets?

-          Line 170, verify the spacing before… and;

-          Line 171 verify the spacing before… to obtain;

-          2.5.3. sub-section, here and in the whole manuscript, decide if to write the common name or the formula of a chemical. The International reader knows the formula of sulphuric acid or other chemical formulas (see line 191);

-          Line 211, verify the spacing;

-          2.6.1. sub-section, a lot of inaccuracies;

-          Line 219: no brackets for 510 nm;

-          2.8.3 sub-section, line 287, do not use this system to indicate the bibliography… by [2]. Be consistent with the guidelines and use some recently published paper as a template;

-          2.8.4 sub-section, line 309: DDPH? Please, verify;

-          Line 319: bracket y/n after a and b? Be consistent;

-          Line 319: no italics for… the;

-          Line 351; verify spacing 4 &5;

-          Line 356: a bracket after b? or no bracket after a? Please, verify;

-          Line 367, verify typo and spacing;

-          Why … &?

-          Line 383, verify the spacing;

-          Table 2 verify typo and size, be consistent with the template;

-          3.3 sub-section, explain that your data are included in table 3;

-          3.3 sub-section, line 470, and in the whole manuscript, do not write in italics the references number [62], be consistent with the instructions for authors;

-          Line 478-479, verify the black color of these lines, it is too faint. Verify the whole manuscript;

-          3.4.1. sub-section, line 489 and in the whole manuscript, when you discuss your data explain the asterisk before mm, or explain in the statistical analysis section;

-          3.5 sub-section, before the discussion of your data, explain that ABTS and DPPH assays are widely applied on many matrices and support this statement with some references such as:

a)      Bergamot (Citrus bergamia, Risso): The Effects of cultivar and harvest date on functional properties of juice and cloudy juice. Antioxidants 8, 221 (2019). DOI:10.3390/antiox8070221

b)      Flavonoids, total phenolics and antioxidant capacity: Comparison between commercial green tea preparations. Food Science and Technology (Campinas) 30(4):1077-1082 (2010). DOI: 10.1590/S0101-20612010000400037

c)       Green-sustainable extraction techniques for the recovery of antioxidant compounds from “citrus Limon” by-products, Journal of Environmental Science and Health, Part B, 57:3, 220-232, DOI: 10.1080/03601234.2022.2046993

-           Captions of figures: Figure and the figure number in bold, please be consistent with the template and verify in some recently published papers;

-          Lines 578 and 579, and in the whole manuscript, the symbols of % immediately after the numeric value: 38.97%;

-          Captions of tables: after the table number you have to insert a dot and not colons;

-          Caption of table 4 (P), statistical analysis section (p), and in the whole manuscript, sometime you have written p in italics and in small letter, sometime no italics and capital letter. Please, be consistent in the whole manuscript;

-          The references section has to be re-arranged to be consistent with the guidelines of Antioxidants: sometime the journal name is missed (ref. 13, 14, 15, 16, 26, 34 and in many others); sometime the journal name is not abbreviated (ref. 19, 74 and in many others); the scientific name in italics (ref 38, 55, 62 and others); and so on, this section has to be completely and carefully re-arranged in light of guidelines;

-          Reference 84, and in the whole section, only the first letter of the author name, in italics;

Author Response

Reviewer 1

Reviewer Comments and Suggestions

Author Response

The manuscript is interesting; however, improvements are necessary.

In the title and in the whole manuscript:  To avoid confusion, please use the correct and updated botanical nomenclature, for example, according to www.gbif.org, and report the authorship and (in brackets) the botanical family in the title and at the first mention.

The authors appreciate the reviewer’s comments and suggestions to improve our article.  As per reviewer comments, we revised accordingly and updated the botanical nomenclature, and highlighted it in the revised manuscript.

Abstract section lines 19, 29, and in the whole manuscript, when you have written the numerical values and the standard deviation, sometimes you have included spaces before and after the symbol ± and sometimes non. Please, be consistent in the whole manuscript, tables, and figures.

As per reviewer comments, we revised our whole numerical data uniformly (lines 19-29), and has been highlighted in yellow color.

Abstract section, line 37, and in the whole manuscript, separate the numeric value from the unity of measurement.

It has been revised as per reviewer comments and it has been highlighted

Introduction section line 57, delete one space after … pain.

Space deleted

Introduction section lines 79-86: the color of these lines is too faint, please, verify.

Corrected and highlighted in the revised manuscript.

Introduction section lines 94-95, please, re-write this sentence because it is not comprehensible and verify verbs.

According to your suggestion lines, 94-95 have been rephrased and highlighted

Introduction section line 98, after ingredients, insert colons instead of a semi-colon.

The colon is inserted and highlighted in line 98.

Line 98, decide if write the name of molecules in small or in capital letters and be consistent in the whole manuscript.

The name of molecules started from small letters and was highlighted in the revised article

Line 111, delete one space before Scattered.

Space deleted

Lines 112-113 and in the whole manuscript, I think it is not necessary to repeat (5) between brackets, the International Reader knows that five is (5); be consistent in the whole manuscript when you have used the same procedure.

Worthy reviewer your suggestion has been addressed in the whole manuscript and highlighted.

Lines 115-116, verify the English form of this sentence, in addition, what is the subject of… prefer to grow?

These words were written for the plant habitat, and suitable conditions for the growth of the plant but rephrased the words as per your suggestion

2.1 sub section, n-Hexane, please decide if small or capital letter: n-hexane or n-Hexane. Be consistent with other chemicals written in your manuscript.

Corrected and choose n-Hexane in the entire article and highlighted

2.1 sub-section line 132 (distilled water), line 136 (deionized water), and in the whole manuscript They are not the same. Which have you used?

Worthy reviewer distilled water was used. The typographical mistake was rectified in the revised version

2.2 sub-section, verify the spacing before and after the hyphen of May –July.

Space deleted

2.3.1 sub-section, line 151, delete one space before… The powder.

Space deleted

Line 152 and in the whole manuscript, when you write a temperature, separate the numeric value from the symbol: 4 °C and not 4°C.

Separated the numeric value from the symbol in the revised article and highlighted

Line 152, replace Twenty-seven hundred (2700) grams … with …. 2700 g.

Replaced with 2700 g and highlighted with yellow color in the revised version.

Line 153: … with 6 L, is enough, correct the whole manuscript.

Worthy reviewer Yes and corrected as per your suggestion in the revised

Line 162, delete one space before MeOH.

Deleted

Lines 163-164, delete one space after SPM and before the second bracket.

Deleted

Line 164, why 640 grams between brackets?

Brackets removed and replaced with 640 g as per your suggestion and highlighted

Line 170, verify the spacing before… and.

Corrected

Line 171 verify the spacing before… to obtain.

Corrected

2.5.3. sub-section, here and in the whole manuscript, decide if to write the common name or the formula of a chemical. The International reader knows the formula of sulphuric acid or other chemical formulas (see line 191);

Worthy reviewer the common names were deleted from the whole manuscript and only kept the chemical formulas in the revised article

Line 211, verify the spacing.

Corrected

2.6.1. sub-section, a lot of inaccuracies.

Corrected as per your suggestion and highlighted

Line 219: no brackets for 510 nm.

Brackets deleted

2.8.3 sub-section, line 287, do not use this system to indicate the bibliography… by [2]. Be consistent with the guidelines and use some recently published papers as a template.

Added and followed recently published articles for keeping the uniform format

2.8.4 sub-section, line 309: DDPH? Please, verify.

Thanks, Worthy reviewer the incorrect DDPH was Replaced with the correct abbreviation DPPH and highlighted

Line 319: bracket y/n after a and b? Be consistent.

As per reviewer comment, revised uniformly in the revised article

Line 319: no italics for… the.

Corrected and highlighted

Line 351; verify spacing 4 &5.

Corrected in the revised version and highlighted

Line 356: a bracket after b? or no bracket after a? Please, verify.

Bracket added for uniform format

Line 367, verify typo and spacing. Why … &?

Corrected and & is short but replaced with and in the revised version and highlighted

Line 383, verify the spacing.

Corrected

Table 2 verify typo and size, and be consistent with the template.

All the tables were made according to the template and recently published article to keep uniformity in the entire article

3.3 sub-section, explain that your data are included in table 3.

Corrected and highlighted in the revised version

3.3 sub-section, line 470, and in the whole manuscript, do not write in italics the reference number [62], be consistent with the instructions for authors.

Corrected and highlighted yellow color.

Line 478-479, verify the black color of these lines, it is too faint. Verify the whole manuscript.

Corrected and checked the whole article

3.4.1. sub-section, line 489, and in the whole manuscript, when you discuss your data explain the asterisk before mm or explain it in the statistical analysis section.

As per your suggestion, the asterisk was removed from the text and explained in the statistical analysis section and the respective part below the tables and figures.

3.5 sub-section, before the discussion of your data, explain that ABTS and DPPH assays are widely applied on many matrices and support this statement with some references such as:

Worthy reviewer, you have proposed some valuable literature that improved this article and will be helpful in our future studies.

a)  Bergamot (Citrus bergamia, Risso): The Effects of cultivar and harvest date on functional properties of juice and cloudy juice. Antioxidants 8, 221 (2019). DOI:10.3390/antiox8070221

The literature has been added and highlighted in the result and discussion section as a reference 95.

b)  Flavonoids, total phenolics, and antioxidant capacity: Comparison between commercial green tea preparations. Food Science and Technology (Campinas) 30(4):1077-1082(2010). DOI: 10.1590/S0101-20612010000400037

The revised version supplemented with the data and highlighted as reference 96

c)    Green-sustainable extraction techniques for the recovery of antioxidant compounds from “citrus Limon” by-products, Journal of Environmental Science and Health, Part B, 57:3, 220-232, DOI:10.1080/03601234.2022.2046993

As per your suggestion and improvement of the manuscript literature has been added and highlighted in the revised version as in the result and discussion and reference section as 94.

Captions of figures: Figure and the figure number in bold, please be consistent with the template and verify in some recently published papers.

Corrected as per followed your suggestion, template, and recently published articles and highlighted

Lines 578 and 579, and in the whole manuscript, the symbols of % immediately after the numeric value: 38.97%.

Corrected in the whole manuscript.

Captions of tables: after the table number you must insert a dot and not colons.

Corrected and highlighted as per your valuable advice

Caption of table 4 (P), statistical analysis section (p), and in the whole manuscript, sometimes you have written p in italics and small letters, sometimes no italics and capital letters. Please, be consistent in the whole manuscript;

Corrected and highlighted and keep one format in the entire article as per your recommendations

The references section must be re-arranged to be consistent with the guidelines of Antioxidants: sometimes the journal name is missed (ref. 13, 14, 15, 16, 26, 34, and in many others); sometimes the journal name is not abbreviated (ref. 19, 74 and in many others); the scientific name in italics (ref 38, 55, 62 and others); and so on, this section has to be completely and carefully re-arranged in light of guidelines.

Corrected as per your suggestion one by one and rearranged followed the template, recently published article and highlighted in the revised manuscript.

Reference 84, and in the whole section, only the first letter of the author's name, is in italics.

Corrected and highlighted in the revised article.

Reviewer 2 Report

The authors concluded that S. petiolata contains bioactive components with many biological properties that may be responsible for several therapeutic effects. All the samples tested, especially the SPM extract, have the highest content of flavonoids and phenols.
The work is generally well written. However, the manuscript must be supplemented with the following information:
1) Were the applied research methods validated?
2) What is the repeatability, precision and accuracy of the determinations?

Author Response

Reviewer 2

Reviewer Comments

Author response

The work is generally well written. However, the manuscript must be supplemented with the following information:

Worthy reviewer thanks for your appreciation and for considering our work

1) Were the applied research methods validated?

Dear Reviewer

We have determined the quantification of total flavonoids, phenols, and antioxidant activity through published protocols using the Uv spectrophotometer to see the quantity and the effect of these constituents on the biological activities and validated them. As this is our ongoing project, that’s why we will further do the quantification of active ingredients through a Uv spectrophotometer which will be validated by another method like HPLC.     

2) What is the repeatability, precision, and accuracy of the determinations?

As a natural product chemist,  we have to isolate some of the active and novel chemical ingredients from the understudy plant for responsible therapeutic significance so here we have only identified the secondary metabolites of the most active extracts based on the observed activities through ESI-LC-MS which does not need these parameters like repeatability, precision, and accuracy. We will determine all these parameters through HPLC analysis in the coming project. 

Round 2

Reviewer 1 Report

As I commented in my previous evaluation, the manuscript is very interesting. In this version, authors have applied all correction and completely replied all comments. In present form the manuscript could be publish.

Author Response

Author response to Ist reviewer

All the authors are thankful to the Worthy reviewer for their appreciation and recommendation of our work for publication in this prestigious journal.
